# Topological and Mechanical Properties of Different Lattice Structures Based on Additive Manufacturing

**DOI:** 10.3390/mi13071017

**Published:** 2022-06-27

**Authors:** Fei Teng, Yongguo Sun, Shuai Guo, Bingwei Gao, Guangbin Yu

**Affiliations:** 1School of Mechanical and Power Engineering, Harbin University of Science and Technology, 52 Xuefu Road, Nangang District, Harbin 150000, China; tengfei@hrbust.edu.cn (F.T.); gaobingwei@hrbust.edu.cn (B.G.); 2Harbin Shipbuilding Boiler and Turbine Research Institute, 35 Honghu Road, Daoli District, Harbin 150010, China; guoshuai@hrbust.edu.cn; 3School of Mechatronics Engineering, Harbin Institute of Technology, 92 West Dazhi Street, Nangang District, Harbin 150000, China; yugbhit@hrbust.edu.cn

**Keywords:** additive manufacturing, numerical simulation, compression experiment, selective laser melting, triply periodic minimal surface (TPMS)

## Abstract

The appearance and development of additive manufacturing technology promotes the production and manufacture of parts with more complex designs and smaller sizes and realizes the complex topology that cannot be made by equal-material manufacturing and submanufacturing. Nowadays, the application of tri-periodic minimal surface (TPMS) in topology optimization design has become a new choice, and, because of its excellent structure and properties, has gradually become mainstream. In this paper, the mechanical properties of four different topologies prepared by selective laser melting (SLM) using 316L stainless steel powder were investigated, including two TPMS sheet structures (Primitive surface, Gyroid surface) and two common lattice structures (Bcc lattice, truss lattice). The mechanical properties (Young’s modulus, yield stress, plateau stress, and toughness) were compared by numerical simulation and compression experiment. It can be concluded from the results that the mechanical properties and deformation mechanism of the specimen are mainly related to the type of lattice, though have little relationship with unit thickness at the same relative density. The Gyroid curved structure showed the best mechanical properties and energy absorption capacity, followed by the truss lattice structure. By comparison, the mechanical properties of the traditional Bcc lattice structure and the Primitive surface structure are poor, and the deformation mechanism of these two structures is uncertain and difficult to control.

## 1. Introduction

The development of additive manufacturing technology [1] began in the 1980s and is considered a subversive manufacturing technology that promotes the third industrial revolution. It adopts the method of “layered printing and layer by layer accumulation” to prepare parts, thus it is called 3D printing. Compared with traditional manufacturing methods (equal-material manufacturing and subtractive manufacturing), additive manufacturing is freer and more imaginative in design and processing. This liberalization of design makes it possible to manufacture lightweight structures and increase functions.

Lightweight aims to minimize the dead weight of the structure under given boundary conditions while meeting certain life and reliability requirements [2]. There are roughly two ways to achieve the goal of lightweight: material weight reduction and structure weight reduction, and pore structure is widely used in aerospace, automobiles, ships, and other fields due to its good mechanical properties, vibration reduction, and energy absorption effect [3,4,5,6,7].

So far, the most common pore structures are honeycomb structures [8,9] and lattice structures [10]. The former has obvious anisotropy characteristics of structure and performance, and the application of the site will be limited, but it is undeniable that the honeycomb structure is an excellent structure; lattice structure is a periodic structure composed of nodes and struts, which has the characteristics of ultra-high porosity and structural stability, and its structure has many varieties, such as the initial truss lattice to the body-centered cubic (Bcc) lattice and face-centered cubic (Fcc) lattice [11], which is very popular in recent years.

In addition to the above-mentioned two kinds of structures, in recent years, some researchers have began to try to apply the special mathematical model of triply periodic minimal surface (TPMS) [12] to the design of pore structures. In 1873, the physicist J. Plateau used the famous soap-film experiment to discover the surface with the smallest area bounded by a given spatial closed curve C, that is, the minimal surface in physical concept. In mathematics, TPMS is a kind of mathematical surface with three-dimensional periodicity, zero curvature, and large surface area. TPMS, as an emerging pore structure, has attracted wide attention and has been applied to many engineering disciplines, such as tissue engineering [13,14] and structural engineering [15,16,17]. They have been proved to have the best thermal conductivity and conductivity [18], the best fluid permeability [19], tunable acoustic attenuation, and transmission [20], and properties of visible photonic crystals [21]. Tom-based structures can either create thin sheet cell structures by thickening minimal surfaces or create skeleton-based cell structures by solidifying volumes surrounded by minimal surfaces [13]. TPMS has high designability and can achieve the innovative design of a three-period minimal surface porous structure by using gradient design and hybrid design. Gyroid, Primitive, and I-WP are the most-studied TPMS.

Fabrication of any pore structure is difficult due to its complex geometry, especially when the unit size is very small [22,23,24]. The emergence and development of additive manufacturing technology have made it possible to produce such complex geometric shapes [25,26,27]. However, the fabrication techniques and process parameters for such structures should be carefully selected, as AM technology has limitations in removing the supporting structures and internal powders. One of the potential process routes for the fabrication of TPMS sheet structures is through AM technology based on powder beds, as powders can partially support suspended structures within allowable construction angles [28]. Selective laser melting (SLM) is one of the most popular and effective powders in bed-based AM technologies [29,30]. This technique uses a laser beam to selectively melt and fuse metal powder according to a 3D computer-aided design (CAD) model, which is very suitable for directly manufacturing various metal TPMS sheet structures with fine characteristics [31,32].

Numerous research tests have been carried out on various pore structures, such as TPMS sheet structures, which show better strength and better energy absorption capacity than traditional lattice structures [33,34]. When the material and relative density are fixed, the mechanical properties of the cell structure depend largely on the structure and unit size of the cell. Researchers put a lot of effort into finding the optimal cell topology and size to maximize the stiffness/strength/weight ratio. Therefore, attention has been paid to the role of cell topology in enhancing mechanical properties [18]. In particular, in the field of tissue engineering, extensive research has been conducted to determine optimal topologies that can mimic the properties of bone and provide a viable environment for the recovery and regeneration of damaged tissue cells [19,20].

Most of the previous studies used the two variables of different structures and different relative densities to conduct experimental studies. Many studies have concluded that the mechanical properties of the same structural materials are stronger with a higher relative density. Four types of pore structures (Gyroid surface, Primitive surface, truss lattice, Bcc lattice) were printed using 316L stainless steel as an SLM fabrication method. The compressive test and explicit finite element analysis were used to compare the mechanical properties of the four structures in a cube of 50 mm × 50 mm × 50 mm with the same relative density. The mechanical properties and deformation mechanism of the four structures were studied, and the influence of different unit sizes on the structural strength of the same structure with the same relative density was explored. Research shows that the Gyroid TPMS structure significantly outperforms the other three in stiffness (Young’s modulus), strength (yield stress, plateau stress), and energy absorption (toughness), despite the printing process bias. The second is the truss lattice structure, which still has excellent mechanical properties. In this paper, the numerical simulation method is used to verify the comparison. Although there are differences between the numerical simulation results and the experimental structure, the differences are small, and the results are in good agreement. The influence of microstructure and material properties on compression behavior provides new insights into the superior energy absorption and mechanical properties of TPMS structures.

## 2. Design and Modeling

It is relatively easy to model the body-center cube structure. Here, the size of the Bcc lattice structure selection unit is 10.2 mm, 12.5 mm, and 15.3 mm, and the unit thickness is 2 mm, 2.45 mm, and 3 mm, respectively.

To reduce stress, the truss lattice structure is optimized to a truss-like lattice structure, which produces a rounded transition at the focal point of the rod. The selection unit size is 7.57 mm, 9.81 mm, 12.61 mm; unit thickness is 2.7 mm, 3.5 mm, 4.5 mm.

The modeling method based on the minimum surface of three periods is to express the characteristics of the structure by mathematical means. The general expression of TPMS is:(1)$$\varphi (r)=\sum_{K=1}^{K}A_{K}\cos\left[\frac{2\pi \left(h_{k}\cdot r\right)}{\lambda_{k}}+P_{k}\right]=C$$

The above formula expresses the surface through implicit function [34]. On the surface expressed, the value of each point is constant, therefore the surface is also called isosurface. Where *A_K_* represents the amplitude of the function, *λ_k_* affects the period of the function, *P_k_* is the phase shift of the function, and the function value *C* controls the offset of the surface. When *C* = 0, the surface defined by the implicit function divides the space into two parts with equal volume, and the volume fraction of the porous structure formed is 50%.

Table 1 lists some typical TPMS functions. The distance constant *C* can adjust the position of the boundary between the pore and the solid material to change the volume fraction of the structure. The volume fraction of the porous structure decreases with the increase in *C* value. When *C* is greater than a certain range, the surface discontinuity will be caused. The decrease in the *C* value will make the solid volume gradually increase. When it is smaller than a certain range, it will lead to interference on the surface, therefore the value of *C* should be kept within a certain range.

The Primitive surface and Gyroid surface are established in this paper. The specific modeling method is used to extract the zero-level set surface from the equation to establish the TPMS without thickness and then evenly offset the curvature in two directions. The offset solid model will produce small errors, and the errors will be eliminated. The specific operation is to use The Grasshopper plug-in in Rhino to establish TPMS, and use the Materialise Magics software to offset, repair, and smooth the marching surface to obtain a perfect entity model. Then, the model is imported into professional slicing software for slicing to prepare for 3D printing. See Figure 1 for the specific modeling process.

In general, four types of structures are discussed: body-centered cubic lattice structure, truss lattice structure, Primitive surface structure, and Gyroid surface structure. Transverse comparison of mechanical properties of different structures. Generally speaking, the difference of structure is the biggest variable affecting mechanical properties, followed by porosity; the lower the porosity, the greater the relative density, the stronger the overall mechanical properties, and then the influence of structure size and thickness on strength. This paper mainly studies the mechanical properties of different structures with the same relative density. Three different sizes of each structure are studied for longitudinal comparison.

The mathematical models and printed solid samples of four different structures are shown in Figure 2. All structures are constrained by the same relative density under the same cube size to ensure the rigor of the experiment. Detailed parameters of the structure studied in this paper are shown in Table 2.

## 3. Printing and Morphological Characteristics

### 3.1. Manufacture

According to the characteristics of the four structures and the metal powder materials used (316L stainless steel), the SLM metal printer (ZRapid Tech SLM280) is equipped with a laser power of 500 W, and a continuous IPG mirror fiber laser is selected to print parts in a nitrogen-filled printing chamber to reduce oxidation. SLM processing parameters are shown in Table 3, and the SLM printer and printing process are shown in Figure 3. The metal powder used in this study is spherical 316L stainless steel powder with an average particle size of 15–53 μm and a loose density of 3.9 g/cm^3^.

The test blocks of the four structures were wire-cut after printing to remove excess support, and then the heat treatment of the four test blocks was carried out in a high-temperature vacuum furnace (KJ-M1750-12LZ-S). Heat treatment at 650 °C for 2 h and then cooling in furnace. This heat treatment helps relieve the thermal stress caused by the high cooling rate in the SLM process [35,36].

### 3.2. Morphology Characterization

Due to the characteristics of TPMS, the cross-sections of the two types of TPMS in the samples did not appear with sharp corners and burrs but rather showed smooth connections and transitions. The truss lattice structure and the body center cube structure also showed a good enough sample shape; due to the printing method of additive manufacturing, the printing slice is fine enough, and the structural modeling is fine enough.

Because the sample size of this test is sufficient, it is easy to observe the printing inside the structure, therefore there is no need for an electron microscope to observe the inside of the test block. No obvious macroscopic defects such as deformation, bubbles, or large cracks were observed, and the geometry of the test block was consistent with the modeling.

The size and relative density of the measured print results are similar to those of the modeling. The size and density of all printed test blocks are slightly smaller than the design size due to residual powder particles attached to the printing platform during the printing process. The specific test block size and design are shown in Table 2. It can be seen from the table that the sizes of the two TPMS wafer structures differ most from those of CAD modeling, which may be related to the surface area of each unit. By measuring the surface area, it is obvious that the TPMS structure has a higher surface area than the other two lattice structures. As the surface area increases, so too does the interaction between the solidified structure and the powder bed, which usually results in the partial melting of the loose powder below the powder layer, especially when scanning the vector sintering contour trajectory, where some powder particles partially melt and join together.

## 4. Finite Element Modeling and Compression Testing

### 4.1. Finite Element Modeling

Before the real test, the results are predicted by finite element analysis and verified with the test results in both directions.

Due to the periodicity of the structure and its deformation, the physical properties of periodic structures such as Young’s modulus, Shear modulus, and Poisson’s ratio can be analyzed on elements with appropriate boundary conditions [37,38]. However, when the deformation process of parts reaches the degree of yield and fracture, there will be a difference between the deformation of the element structure and the whole structure. It is necessary to carry out finite element analysis of a full-size model rather than an analysis of element structure to accurately analyze the whole process period of the compressive deformation of structural test blocks. Therefore, finite element analysis is carried out on the full-size models of the four structures to ensure that the simulation conditions are consistent with the experiment.

The commercial finite element analysis software ANSYS Workbench 2020R2 was used to carry out the numerical simulation of four kinds of structural blocks to capture the compression response and failure behavior of each structure. The isotropic hardened, molded 316L stainless steel in Workbench is selected as the material. The simulation here does not include a failure mechanism because the fracture and failure are not obvious at the beginning of the deformation.

A tetrahedral mesh with an average size of 0.5 mm (Solid187) was used to model the four structures. Before loading boundary conditions, to ensure that the simulation process was as similar as possible to the actual test, endplates with minimal thickness are set at the top and bottom of the model to make it easier to apply force during simulation without affecting the overall mechanical properties of the structure as much as possible. At the same loading rate as the experiment, the model was fixed by applying a uniform load on the roof and full constraint on the bottom plate. Hide the top and bottom endplates in the resulting stress cloud.

### 4.2. Compression Testing

The compression test was carried out on an INSTRON 3328 universal mechanical testing machine, and the loading direction was the same as the printing direction of the parts [39]. The universal mechanical testing machine and test process are shown in Figure 4. The compression deformation rate is 9 × 10^−3^ mm/s. Linear variable differential transformer (LVDT) was used to collect deformation data, and the collapse of each specimen was recorded by video at a 50 Hz frame rate.

### 4.3. Simulation and Experimental Results

Figure 5 shows the stress–strain curves of the four structures after compression tests (solid lines for each color in the figure). It can be seen from the figure that the stress–strain curves of the four structures all show four different regions: first, the stress increases linearly with the strain, which is in the linear elastic range, and the slope of the curve is Young’s modulus; the second region begins when the stress–strain curve deviates from the linear response and enters a nonlinear state, and the stress increases continuously until the maximum stress peak is reached. When the stress reaches the peak, the sample begins to appear to have an obvious deformation, which is generally shown as an arc-shaped curve. The third zone is the yield stage, where the deformation of the part continues until the structure completely fails. At this time, the curve shows a fluctuation or smooth transition, and the strain keeps increasing while the stress changes very little. If the force continues to be applied, the parts will be compressed. At this time, the fourth stage is the failure stage. After the continuous deformation in the yield stage, the test block begins to densify, and the stress–strain curve shows a sharp rise. The experimental data in Figure 5 also shows the simulation curves of the four structures (dotted lines in the figure). It can be seen from the figure that the stress–strain curve predicted by the numerical method is in good agreement with the experimental results. In the fourth stage of the curve, the overall trend of simulation is quite different from the experimental data because the failure mechanism is not added in the finite element analysis. It should be pointed out that the overall trend of the numerical prediction curve is lower than the experimental result, which is because the external factors of the experiment are uncontrollable and the external conditions of the experiment cannot be as absolute as the numerical simulation; however, this also proves that the external conditions of all experiments are consistent.

As can be seen from the figure, peak stress occurs at the end of the line elasticity of the Primitive curved structure and enters the yield stage, followed by softening and stress fluctuation. Specifically, a stress decrease occurs after the stress enters the yield stage, and then the stress fluctuates back until the densification phenomenon occurs. The Bcc lattice structure shows slight stress fluctuation at the yield stage. In contrast, the other two structures did not show an obvious stress reduction phenomenon at the yield without obvious softening and stress fluctuation phenomenon, and the curve always showed arc-shaped growth. It can also be seen from the figure that test blocks with the same structure show similar mechanical responses, and the overall trend of stress–strain curves depends on different structures but has little relation with unit thickness when the relative Figure 6 shows the stress cloud of the numerical simulation. It is found that the deformation characteristics of the structure are mainly related to the different shapes of the structure and are almost unrelated to the unit thickness under the same relative density. The Primitive curved surface structure and Bcc lattice structure were subjected to oblique shear strain after yielding, which resulted in deformation and collapse. The corresponding stress characteristics could also be seen in the stress cloud of numerical simulation. Figure 7 shows the experimental compression deformation of the Primitive surface structure and Bcc lattice structure. As can be seen from the figure, due to the action of shear stress, the test block presents irregular oblique strain, which makes the deformation uncontrollable. The stress–strain curves show softening and fluctuation. The deformation and failure of the corresponding Gyroid curved structure and truss lattice structure is continuous layer by layer, which shows continuous hardening in the stress–strain curve, which shows continuous arc-shaped growth. The deformation begins closer to the center of the structure and spreads upward and downward.

The physical concept of toughness is used here to try to determine the energy absorption capacity of these four structures. Toughness refers to the ability to absorb energy during plastic deformation and fracture. The better toughness is, the less possibility of brittle fracture occurs, that is, the more energy is absorbed during deformation and before fracture.

The mechanical properties of all test blocks were extracted from stress–strain data in numerical simulation and compression experiments. Here, the mechanical properties of each test block were judged mainly based on Young’s modulus, yield stress, plateau stress, and toughness. The Young’s modulus is calculated according to the slope of the curve at the linear elastic stage, the yield stress is calculated according to the obvious inflection point when the curve enters the yield stage, the plateau stress is calculated according to the average stress within the range of 0.2–0.3 strain, and the toughness is the area under the stress–strain curve within the range of the 0–0.25 strain. Figure 8 shows the elastic modulus, yield stress, platform stress, and toughness of the four structures as a function of unit thickness. Since the relative density of this experiment is fixed, we do not make any processing of the data, and directly compare the four mechanical properties.

It can be seen that the main factor affecting the mechanical properties is the difference of structure, and, under the premise of fixed relative density, the influence of unit thickness on the mechanical properties is very small. Among the four kinds of structures, the mechanical properties of the Gyroid surface structure are better than the other three kinds of structures.

## 5. Conclusions

In summary, the compressive response of four different structures based on additive manufacturing using 316L stainless steel powder as the material was studied by numerical simulation and compression experiment. The research carrier is two triply periodic minimal surface structures (Gyroid surface and Primitive surface) and two common topological structures (Bcc lattice structure and truss lattice structure). The mechanical properties and energy absorption of the four structures are compared by Young’s modulus, yield stress, platform stress, and toughness. The main conclusions are as follows:The precision camera and naked eye observation show that all test blocks printed by SLM technology are in good shape without obvious macro defects, such as deformation, bubbles, or large cracks. The printed test blocks have a high degree of reduction;Under unidirectional compression test and numerical simulation, a continuous hardening of the Gyroid curved and truss lattice structures on the stress–strain curves is observed, and the stress softening and fluctuation are negligible;Due to shear stress, the surface structure of Primitive showed obvious softening and stress fluctuation on the stress–strain curve. The Bcc lattice structure also shows slight stress fluctuations on the curve due to the yield failure of its rigid rods;The result of numerical simulation has a high coincidence with the structure of the compression experiment. The difference between the two mainly lies in the fact that the printing process cannot achieve the perfect reduction of the structure, and that the external conditions of the compression experiment are changeable, therefore all conditions cannot be taken into account in the simulation;The Gyroid TPMS significantly outperforms the other three in stiffness (Young’s modulus), strength (yield stress, plateau stress), and energy absorption (toughness). The second is the truss lattice structure, which still has excellent mechanical properties;Through the exploration of the mechanical properties of the four structures, the superiority of the TPMS structure can be concluded, which can be used as a reference for future topological structure research.

## Figures and Tables

**Figure 1 micromachines-13-01017-f001:**
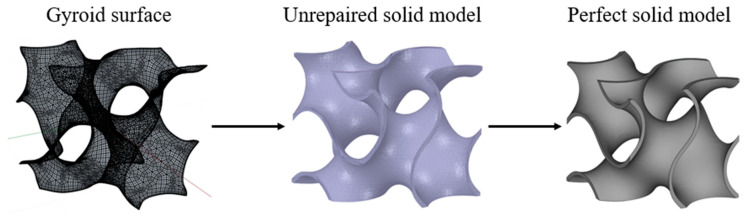
The concrete modeling process of minimal surface.

**Figure 2 micromachines-13-01017-f002:**
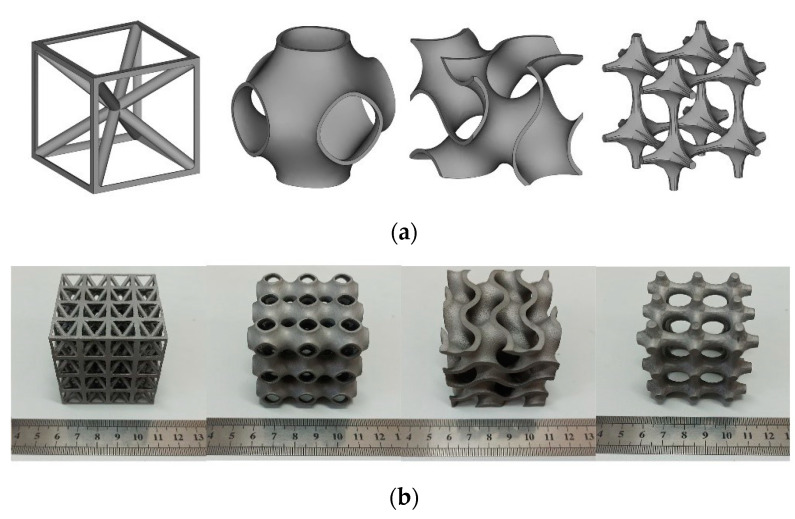
Structure diagram of each unit (**a**) and actual printed solid parts (**b**), from left to right, there are Bcc lattice, Primitive, Gyroid, and truss lattice.

**Figure 3 micromachines-13-01017-f003:**
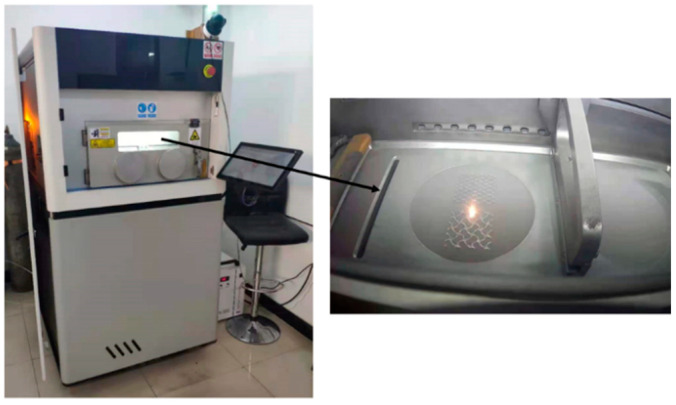
SLM printer and the printing process.

**Figure 4 micromachines-13-01017-f004:**
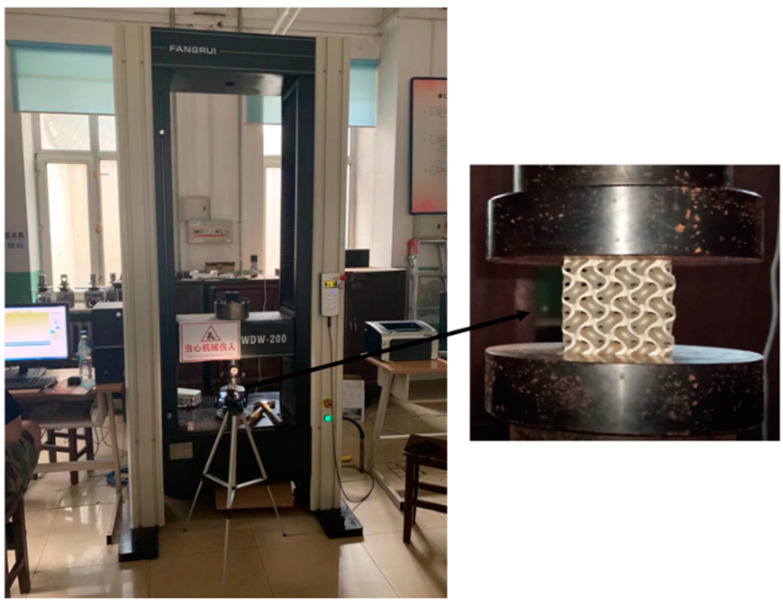
Universal mechanical testing machine and test process.

**Figure 5 micromachines-13-01017-f005:**
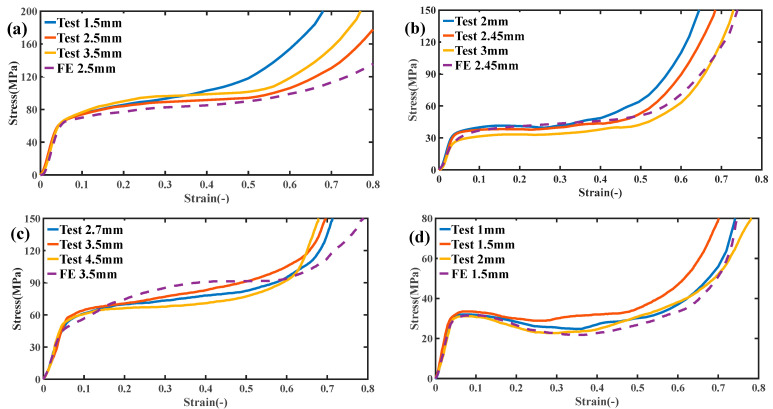
Experimental and numerical simulation of stress–strain curves for (**a**) Gyroid surface, (**b**) Bcc lattice structure, (**c**) Primitive surface, (**d**) truss lattice structure.

**Figure 6 micromachines-13-01017-f006:**
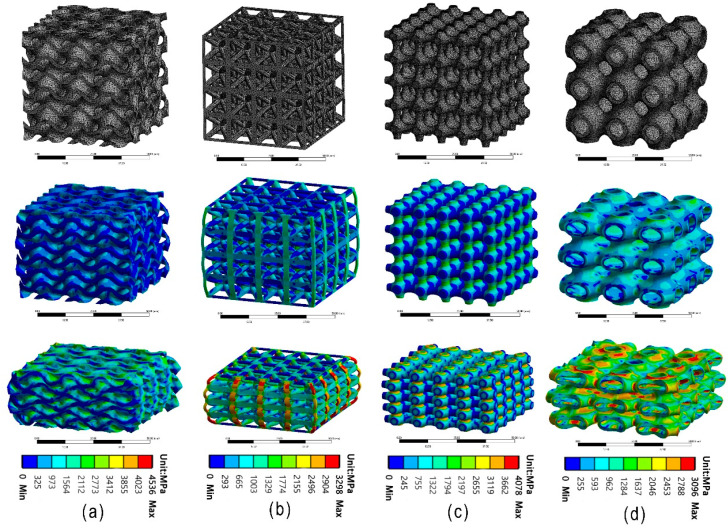
(**a**) Gyroid surface, (**b**) Bcc lattice structure, (**c**) Primitive surface, (**d**) truss lattice structure, meshing and stress cloud of finite element analysis under given strain.

**Figure 7 micromachines-13-01017-f007:**
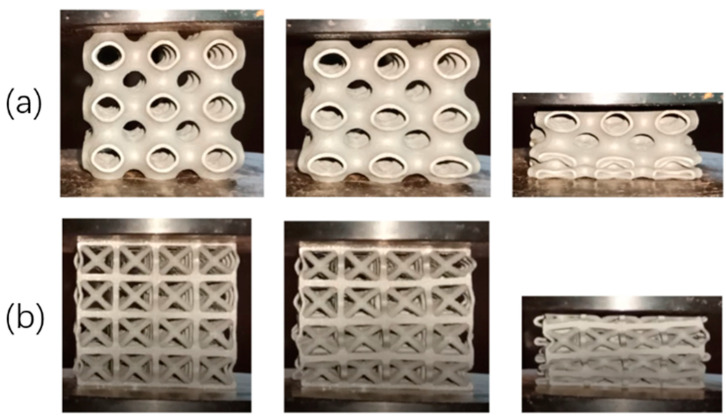
(**a**) Primitive surface, (**b**) Bcc lattice structure, deformation under fixed strain.

**Figure 8 micromachines-13-01017-f008:**
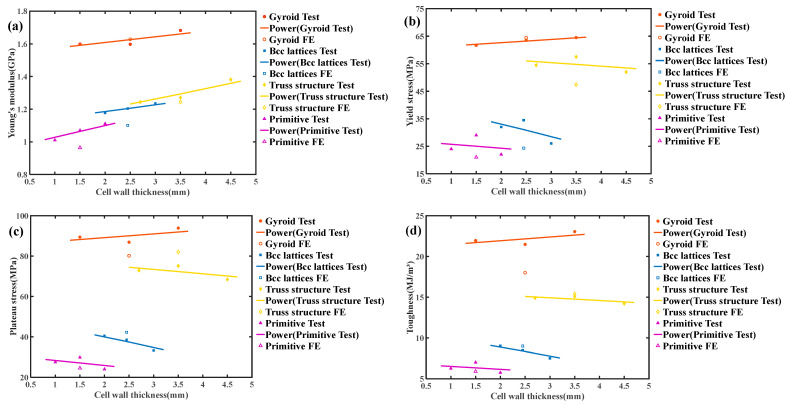
Normalized mechanical properties with respect to the measured relative density: (**a**) Young’s modulus, (**b**) yield stress, (**c**) plateau stress, (**d**) toughness. Solid lines indicate the best fitting lines of experimental data using power laws.

**Table 1 micromachines-13-01017-t001:** Typical TPMS functions.

Name	TPMS Function	Unit Interval
Primitive	$\Phi_{P}(x,y,z)=\cos(x)+\cos(y)+\cos(z)=c$	(0, 2π)
Gyroid	$\Phi_{G}(x,y,z)=\cos(x)\sin(y)+\cos(y)\sin(z)+\cos(z)\sin(x)=c$	(0, 2π)
Diamond	$\begin{array}{ll} \Phi_{D}(x,y,z)= & \sin(x)\sin(y)\sin(z)+\cos(x)\sin(y)\sin(z)+\\ & \sin(x)\cos(y)\sin(z)+\sin(x)\cos(y)\cos(z)=c \end{array}$	(−π, π)
I-WP	$\Phi_{I}(x,y,z)=\cos(x)\cos(y)+\cos(y)\cos(z)+\cos(z)\cos(x)-\cos(x)\cos(y)\cos(z)=c$	(0, 2π)
Neovius	$\Phi_{N}(x,y,z)=3(\cos(x)+\cos(y)+\cos(z))+4\cos(x)\cos(y)\cos(z)$	(0, 2π)

**Table 2 micromachines-13-01017-t002:** The specific parameters of the structure used in this study.

Structure	Nominal Shell Thickness (mm)	Measured Shell Thickness (mm)	Nominal Cell Size (mm)	Nominal Relative Density [-]	Measured Relative Density [-]
Bcc lattice	2	1.94 ± 0.01	10.2	0.3	0.295 ± 0.000
	2.45	2.35 ± 0.01	12.5	0.3	0.293 ± 0.001
	3	2.95 ± 0.01	15.3	0.3	0.288 ± 0.001
Primitive	1	0.89 ± 0.01	15.2	0.3	0.289 ± 0.001
	1.5	1.38 ± 0.01	17.5	0.3	0.290 ± 0.003
	2	1.91 ± 0.01	19.2	0.3	0.288 ± 0.001
Gyroid	1.5	1.40 ± 0.01	15.4	0.3	0.283 ± 0.000
	2.5	2.39 ± 0.01	25.7	0.3	0.279 ± 0.001
	3.5	3.38 ± 0.01	36.1	0.3	0.291 ± 0.001
Truss lattice	2.7	2.66 ± 0.01	7.6	0.3	0.298 ± 0.003
	3.5	3.42 ± 0.01	9.8	0.3	0.295 ± 0.001
	4.5	4.47 ± 0.01	12.6	0.3	0.295 ± 0.001

**Table 3 micromachines-13-01017-t003:** SLM process parameters.

Parameter	Value
Laser power	500 W
Laser type	Fiber Laser (IPG)
Powder spreading method	Scraper powder spreading
Layer thickness	0.03 mm
Scan speed	2000 mm/s
Spot size	0.05~0.15 mm
Ambient temperature	20~26 °C
Hatch spacing	0.1 mm

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
