# Peer review of "Topological and Mechanical Properties of Different Lattice Structures Based on Additive Manufacturing"

_micromachines, 2022, doi:10.3390/mi13071017_

Round 1
Reviewer 1 Report
The topic of this manuscript is interesting and the manuscript is well organized. The discussion is well done and supported by considerations derived from experimental data and numerical simulation results. Nevertheless, the following minor remarks need to be taken into account:
- an integration of the literature review should be done. It is not exhaustive, references from the last three years being missing.
- clarify the choice of the parameters shown in table 2 (nominal shell thickness and cell size are due to process or experimental constraints, to specific considerations, etc?)
- check for spelling and typing errors (e.g. subscript, uppercase/lowercase, incomplete sentence in the conclusion line 362)
- check figure numbering (figure 1 instead of 5 is shown in the caption of page 9)
- in the caption of figure 6 (page 9) “experimental and numerical simulation” is written but the figure concerns only the numerical simulation
- figure 2 instead of 7 is shown in the caption of page 10 and the caption refers only to the upper part of the page, not also to the mechanical graphs
Author Response
Thank you very much for your comments. For each opinion, I make modifications or explanations below:
1. I have added and adjusted some references to the uploaded manuscript
2.First, select the shell thickness, and then control the relative density to a fixed 0.3, so as to obtain the cell size. Secondly, this paper wants to study whether the shell thickness has a significant impact on the mechanical properties of the test blocks, so three sizes of each type are selected under the conditions of experiment and printing process. The limitations of the printing process are indeed considered here, and only three sizes can be selected in equal proportion within the printable range as far as possible.
3.Modified in manuscript
4.Modified in manuscript
5.Modified in manuscript
6.Modified in manuscript
Please see the attachment.
Thank you again for your comments

Reviewer 2 Report
In this manuscript, the authors have presented case examples how to different lattice structure characteristics changes with respect to geometrical parameters. The study is well done and reported appropriately. Manuscript is easy to read and interestingly written. Results follows basic assumptions. Nice piece of work. I will recommend to publish this paper in the journal.
Author Response
Thank you very much for your review
Reviewer 3 Report
The authors investigated four types of topologically optimised structures based on a triply-periodic minimal surface and fabricated them from 316L stainless steel powder by selective laser melting. They observed their practical compressive responses along with numerical simulation comparison. However, some aspects need improvement or clarification, and in this sense, I would like to make the following comments to improve the overall quality of the work.
1. Figure 2 consists of several panels, which should therefore be listed as (a), (b), etc. with their descriptions. It would be better to provide the names of each structure in the figure.
2. Hatch spacing is also an important manufacturing parameter. It would be good to include it in Table 3.
3. Fig. 5 is not present in the paper.
4. "Fig. 7 shows the experimental compression deformation of the Primitive surface structure and Bcc lattice structure" - Is it possible to show the same for the gyroid and truss lattice structure?
5. The distances between two consecutive figures are the same on all sides of the figures in Figure 6. The panels should be separated using marking.
6. The captions of the figures in the page 10 should be placed in the correct location and the figure number should be changed on the same page. It is also recommended that the panels of the figures be clearly marked.
Author Response
Thank you very much for your comments. For each opinion, I make modifications or explanations below:
1.Modified in manuscript
2."Hatch spacing" has been added.
3.Modified in manuscript
4.This is my description in my paper: "Fig. 7 shows the experimental compression deformation of the Primitive surface structure and Bcc lattice structure. As can be seen from the figure, due to the action of shear stress, the test block presents irregular oblique strain, which makes the deformation uncontrollable. The stress-strain curves show softening and fluctuation. The deformation and failure of the corresponding Gyroid curved structure and truss lattice structure is continuous layer by layer, which shows contin-uous hardening in the stress-strain curve, which shows continuous arc-shaped growth. The deformation begins closer to the center of the structure and spreads upward and downward."
My idea is that the primitive surface structure and BCC lattice structure have special deformation mechanism, which is worth showing in the paper in the form of pictures, while the deformation mechanism of gyroid and truss lattice structure is continuous layer by layer, so there is no need to add more space to reflect it. If necessary, I can also add this part of the picture in the next modification.
5. I think your opinion is very pertinent, but I think the layout here will be more clear.
6.Modified in manuscript
Please see the attachment.
Thank you again for your comments
